

# Plastid genomics of *Nicotiana* (Solanaceae): insights into molecular evolution, positive selection and the origin of the maternal genome of Aztec tobacco (*Nicotiana rustica*)

Furrukh Mehmood[1,2], Abdullah[1], Zartasha Ubaid[1], Iram Shahzadi[1], Ibrar Ahmed[3], Mohammad Tahir Waheed[1], Peter Poczai[2] and Bushra Mirza[1]

[1] Department of Biochemistry, Faculty of Biological Sciences, Quaid-i-Azam University, Islamabad, Pakistan
[2] Botany Unit, Finnish Museum of Natural History, University of Helsinki, Helsinki, Finland
[3] Alpha Genomics Private Limited, Islamabad, Pakistan

Corresponding authors
Peter Poczai, peter.poczai@helsinki.fi
Bushra Mirza,
bushramirza@qau.edu.pk

## ABSTRACT

Species of the genus *Nicotiana* (Solanaceae), commonly referred to as tobacco plants, are often cultivated as non-food crops and garden ornamentals. In addition to the worldwide production of tobacco leaves, they are also used as evolutionary model systems due to their complex development history tangled by polyploidy and hybridization. Here, we assembled the plastid genomes of five tobacco species: *N. knightiana, N. rustica, N. paniculata, N. obtusifolia* and *N. glauca*. De novo assembled tobacco plastid genomes had the typical quadripartite structure, consisting of a pair of inverted repeat (IR) regions (25,323–25,369 bp each) separated by a large single-copy (LSC) region (86,510–86,716 bp) and a small single-copy (SSC) region (18,441–18,555 bp). Comparative analyses of *Nicotiana* plastid genomes with currently available Solanaceae genome sequences showed similar GC and gene content, codon usage, simple sequence and oligonucleotide repeats, RNA editing sites, and substitutions. We identified 20 highly polymorphic regions, mostly belonging to intergenic spacer regions (IGS), which could be suitable for the development of robust and cost-effective markers for inferring the phylogeny of the genus *Nicotiana* and family Solanaceae. Our comparative plastid genome analysis revealed that the maternal parent of the tetraploid *N. rustica* was the common ancestor of *N. paniculata* and *N. knightiana*, and the later species is more closely related to *N. rustica*. Relaxed molecular clock analyses estimated the speciation event between *N. rustica* and *N. knightiana* appeared 0.56 Ma (HPD 0.65–0.46). Biogeographical analysis supported a south-to-north range expansion and diversification for *N. rustica* and related species, where *N. undulata* and *N. paniculata* evolved in North/Central Peru, while *N. rustica* developed in Southern Peru and separated from *N. knightiana,* which adapted to the Southern coastal climatic regimes. We further inspected selective pressure on protein-coding genes among tobacco species to determine if this adaptation process affected the evolution of plastid genes. These analyses indicate that four genes involved in different plastid functions, including DNA replication (*rpo*A) and photosynthesis (*atp*B, *ndh*D and *ndh*F), came under positive selective pressure as a result of specific environmental conditions. Genetic mutations

in these genes might have contributed to better survival and superior adaptations during the evolutionary history of tobacco species.

## INTRODUCTION

*Nicotiana* L. is the fifth largest genus in the megadiverse plant family Solanaceae, comprising 75 species (*Olmstead et al., 2008*; *Olmstead & Bohs, 2007*), which were subdivided into three subgenera and fourteen sections by *Goodspeed (1954)*. The subgenera of *Nicotiana,* as proposed by *Goodspeed (1954)*, were not monophyletic (*Aoki & Ito, 2000*; *Chase et al., 2003*), but most of Goodspeed's sections were natural groups. The formal classification of the genus has been refined to reflect the growing body of evidence that *Nicotiana* consists of 13 sections (*Knapp, Chase & Clarkson, 2004*). One significant utilization of *Nicotiana* species has been as a source of genetic diversity for improving one of the most widely cultivated non-food crops, common tobacco (*N. tabacum* L.). This species is of major economic interest and is grown worldwide for its leaves used in the manufacture of cigars, cigarettes, pipe tobacco, and smokeless tobacco products consumed by more than one billion people globally (*Lewis, 2011*; *Occhialini et al., 2016*). While *N. tabacum* is the most notable commercial species, several additional species are also cultivated for smoking (*N. rustica* L.) and ornamental (*N. sylvestris* Spegazzini & Comes) or industrial (*N. glauca* Graham) purposes (*Lester & Hawkes, 2001*). Aztec or Indian tobacco (*N. rustica*), characterized by short yellowish flowers and round leaves, is widely cultivated in Mexico and North America. It was the first tobacco species introduced to Europe in the 16th century, but later superseded by *N. tabacum* for its milder taste (*Shaw, 1960*). Known as ''*o-yen'-kwa hon'we*'' (real tobacco) by North American Iroquois (*Kell, 1966*), it was used for medicinal and ritual purposes or even in weather forecasting (*Winter, 2001*). Aztec tobacco is still cultivated in South America, Turkey, Russia and Vietnam due to its tolerance to adverse climatic conditions (*Sierro et al., 2018*).

Some members of *Nicotiana* offer several research advantages, including extensive phenotypic diversity, amenability to controlled hybridizations and ploidy manipulations, high fecundity, and excellent response to tissue culture (*Lewis, 2011*). Consequently, *N. tabacum* and *N. benthamiana* Domin have become model organisms in the generation of new knowledge related to hybridization, cytogenetics, and polyploid evolution (*Goodin et al., 2008*; *Zhang et al., 2011*; *Bally et al., 2018*; *Schiavinato et al., 2019*). The first complete plastome nucleotide sequence was published in 1986 for *N. tabacum* (*Shinozaki et al., 1986*). Since then, the structure and composition of plastid genomes has become widely utilized in identifying unique genetic changes and evolutionary relationships of various groups of plants. Furthermore, plastid genes have also been linked with important crop traits such as yield and resistance to pests and pathogens (*Jin & Daniell, 2015*).

Chloroplasts (cp) are large, double-membrane organelles with a genome size of 75–250 kb (*Palmer, 1985*). Most chloroplast encoded proteins are responsible for photosynthesis and for the synthesis of fatty acids and amino acids (*Cooper, 2000*). Angiosperm plastid genomes commonly contain ~130 genes, comprised of up to 80 protein-coding, 30 transfer RNA (tRNA), and four ribosomal RNA (rRNA) genes (*Daniell et al., 2016*). The plastid genome exists in circular and linear forms (*Oldenburg & Bendich, 2015*) and the percentage of each form varies within plant cells (*Oldenburg & Bendich, 2016*). Circular-plastid genomes typically have a quadripartite structure, consisting of two inverted repeat regions (IRa and IRb), separated by one large single-copy (LSC) and one small single-copy (SSC) region (*Palmer, 1985*; *Amiryousefi, Hyvönen & Poczai, 2018a*; *Abdullah et al., 2019a*). Numerous mutation events occur in plastid genomes, including variations in tandem repeats, insertions/deletions (indels), point mutations , while inversions and translocations are also common (*Jheng et al., 2012*; *Xu et al., 2015*; *Abdullah et al., 2020*).

The plastid genome of angiosperms has maternal inheritance (*Daniell, 2007*), which together with its conserved organization makes it extremely useful for exploring phylogenetic relationships at different taxonomic levels (*Ravi et al., 2008*). Plastid genome polymorphisms are useful for species barcoding, solving taxonomic issues, studying population genetics, and for investigating species adaptation to their natural habitats (*Ahmed, 2014*; *Daniell et al., 2016*; *Nguyen et al., 2017*). Genes in the plastid genome encode proteins and several types of RNA molecules, which play a vital role in functional plant metabolism, and can consequently undergo selective pressures. Most plastid protein-coding genes are under negative or purifying selection to maintain their function, while positive selection might act on some genes in response to environmental changes (*Iram et al., 2019*; *Henriquez et al., 2020*).

*Nicotiana* species are diploid ($2n = 2x = 24$), although allopolyploid species are also common in the genus (*Leitch et al., 2008*). Phylogenetic studies have shown that these allopolyploids were formed 0.4 million (*N. rustica* and *N. tabacum*) (*Clarkson, Dodsworth & Chase, 2017*) to 5 million years ago (species of sect. *Suaveolentes*) (*Schiavinato et al., 2020*). *Nicotiana tabacum* ($2n = 4x = 48$), is known to be a natural allopolyploid derived from two closely related ancestors (*Lim et al., 2007*). The paternal donor *N. tomentosiformis* L. ($2n = 24$) was confirmed by genomic *in situ* hybridization (GISH) (*Clarkson et al., 2005*), physical mapping (*Bindler et al., 2011*) and genome sequencing (*Sierro et al., 2014*), while the maternal donor *N. sylvestris* ($2n = 24$) was identified by plastid genome sequencing (*Yukawa, Tsudzuki & Sugiura, 2006*). Aztec tobacco (*N. rustica*), like *N. tabacum,* is also an allotetraploid but has originated from the recent hybridization of different parental species. Based on morphology, karyotype analyses and crossing experiments, *Goodspeed (1954)* suggested the ancestral species are *N. paniculata* L. ($2n = 2x = 24$; maternal) and *N. undulata* Ruiz & Pav. ( $2n = 2x = 24$; paternal). The identity of the parental species was investigated using nuclear internal transcribed spacer (ITS) regions, chloroplast markers *in situ* hybridization methods, and genome sequencing (*Aoki & Ito, 2000*; *Chase et al., 2003*; *Clarkson et al., 2004*; *Lim et al., 2004*; *Lim et al., 2007*; *Kovarik et al., 2004*; *Sierro et al., 2018*). These analyses confirmed *N. undulata* as the paternal ancestor according to Goodspeed's hypothesis and showed the genome of *N. rustica* lacks inter-genomic

translocations (*Lim et al., 2004*; *Kovarik et al., 2012*), while additivity can be observed in the 5S and 35S rDNA loci respect to its progenitors (*Lim et al., 2007*), which were homogenized by concerted evolution (*Kovarik et al., 2004*). These analyses did not provide further evidence for the maternal parent of *N. rustica* but suggested either *N. knightiana* L. or *N. paniculata* could be the maternal donor.

Here, we assembled the plastid genome of five *Nicotiana* species and compared their sequences to gain insight into the plastid genome structure of genus *Nicotiana*. We also inferred the phylogenetic relationship of the genus and investigated the selection pressures acting on protein-coding genes. We then identified mutational hotspots in the *Nicotiana* plastid genome that might be used for the development of robust and cost-effective markers in crop breeding or taxonomy. Lastly, we used this information to trace the origin of the maternal genome of the allopolyploid *Nicotiana rustica*.

## MATERIALS AND METHODS

### Plastid genome assembly and annotation

Illumina sequence data of *N. knightiana* (13.1 Gb, accession number SRR8169719), *N. rustica* (15.5 Gb, SRR8173839), *N. paniculata* (35.1 Gb, SRR8173256), *N. obtusifolia* (23 Gb, SRR3592445) and *N. glauca* (12.5 Gb, SRR6320052) were downloaded from the Sequence Read Archive (SRA). The plastid genome sequence reads were selected by performing BWA-MEM mapping with default settings (*Li & Durbin, 2009*) using *Nicotiana tabacum* (GenBank accession number: NC_001879) as a reference. Geneious R8.1 (*Kearse et al., 2012*) *de novo* assembler was used to order the selected contigs for final assembly by selecting option "Medium sensitivity/Fast", while keeping other parameters as default. Gene annotation was conducted using GeSeq (*Tillich et al., 2017*) with a BLAT (*Kent, 2002*) search of 85% to annotate protein-coding genes, rRNAs and tRNAs; CPGAVAS2 was used with default parameters by selecting option 1 "43-plastome" (*Shi et al., 2019*). After automatic annotation, start/stop codons and the position of introns were further confirmed manually by visual inspection of the translated protein of each gene in Geneious R8.1 and BLAST search using default settings with homologous genes of plastid genomes of Solanaceae. The tRNA genes were further verified by tRNAscan-SE v2.0 with default settings using options: sequence source "Organellar tRNA", search mode "Default", genetic code "Universal", and Cut-off score for reporting tRNAs "15" (*Lowe & Chan, 2016*); ARAGORN v1.2.38 was used with default parameters by selecting genetic code "Bacterial/Plant chloroplast" with maximum intron length of 3,000-bp (*Laslett & Canback, 2004*). Circular genome maps were drawn with OGDRAW v1.3.1 (*Greiner, Lehwark & Bock, 2019*) by uploading the GenBank (.gb) format of each plastid genome and selecting options: "Circular", "Plastid", "Tidy up annotation", and "Draw GC graph". The average coverage depth of *Nicotiana* species plastid genomes was calculated by mapping all raw reads without trimming to *de novo* assembled plastid genomes with BWA-MEM (*Li & Durbin, 2009*) visualized in Tablet (*Milne et al., 2009*). Novel *Nicotiana* plastid genomes were deposited in National Center for Biotechnology Information (NCBI) (Table S1).

## Comparative genome analysis and RNA editing prediction

Novel plastid genome sequences were compared through multiple alignments using MAFFT v7 (*Katoh & Standley, 2013*). All parts of the genome, including intergenic spacer regions (IGS), introns, protein-coding genes, and ribosomal RNAs and tRNAs, were considered for comparison. Each part was extracted and used to determine nucleotide diversity in DnaSP v6 (*Rozas et al., 2017*). Substitutions, transition and transversion rates were also determined compared to the *N. tabacum* reference using Geneious R8.1. Structural units of the plastid genome (LSC, SSC and IR) were individually aligned to determine the rate of substitutions and to further search for indels using DnaSP v6. Structural borders of plastid genomes were compared for 10 selected *Nicotiana* species using IRscope with option "GB file upload" and default settings (*Amiryousefi, Hyvönen & Poczai, 2018b*). The online software PREP-cp (Putative RNA Editing Predictor of Chloroplast) was used with default settings to determine putative RNA editing sites (*Mower, 2009*). Codon usage and amino acid frequencies were determined by Geneious R8.1.

## Repeats analyses

Microsatellites within the plastid genomes of five *Nicotiana* species were identified using MISA (*Beier et al., 2017*) with a minimal repeat number of 7 for mononucleotide repeats, 4 for dinucleotide repeats and 3 for tri-, tetra-, penta- and hexanucleotide SSRs. We also used REPuter (*Kurtz et al., 2001*) with minimal repeat size set to 30 bp, Hamming distance set to 3, minimum similarity percentage of two repeat copies up to 90%, and a maximum computed repeat of 500 for scanning and visualizing forward (F), reverse (R), palindromic (P) and complementary (C) repeats. Tandem repeats were found with the Tandem Repeats Finder using default parameters (*Benson, 1999*).

## Non-synonymous ($K_a$) and synonymous ($K_s$) substitution rate analysis

To determin $K_a$ and $K_s$, protein-coding genes were extracted from the newly assembled *Nicotiana* plastid genomes and aligned using MAFFT with the corresponding genes of the previously published plastid genome of *N. tabacum* (Z00044.2) as a reference and analyzed using DnaSP v6. The data were interpreted in terms of purifying selection ($K_a/K_s < 1$), neutral evolution ($K_a/K_s = 1$), and positive selection ($K_a/K_s > 1$).

We evaluated the impact of positive selection using additional codon models to estimate the rates of synonymous and nonsynonymous substitution. The signs of positive selection were further assessed using fast unconstrained Bayesian approximation (FUBAR) (*Murrell et al., 2013*) and the mixed effects model of evolution (MEME) (*Murrell et al., 2012*) as implemented in the DATAMONKEY web server (*Delport et al., 2010*). Sites with cut-off values of PP > 0.9 in FUBAR were considered as candidates to have evolved under positive selection. From all the analyses performed in DATAMONKEY, the most suited model of evolution for each dataset was selected as directly estimated on this web server. In addition, the mixed effects model of evolution (MEME), a branch-site method incorporated in the DATAMONKEY server, was used to test for both pervasive and episodic diversifying selection. MEME applies models with variable $\omega$ across lineages at individual sites,

restricting $\omega$ to $\leq 1$ in a proportion p of branches and unrestricted at a proportion (1 − p) of branches per site. Positive selection was inferred with this method for *p* values < 0.05 using the false discovery rate (FDR) correction according to *Benjamini & Hochberg (1995)* in Microsoft Excel.

## Phylogenetic analyses

Plastid genome sequences of the genus *Nicotiana* were selected from the Organelle Genome Resources of the NCBI (accessed on 21.02.2019) and used in phylogenetic inference along with *de novo* assembled sequences of *Nicotiana*. The $x = 12$ clade includes the traditional subfamily Solanoideae plus *Nicotiana* with the Australian endemic Anthocercideae tribe and takes its name from the synapomorphy of chromosome numbers based on 12 pairs (*Olmstead et al., 2008*). *Nicotiana* and Anthocercideae appear to be in a first branching position in the $x = 12$ clade, thus we have chosen *Solanum dulcamara* L. (*Amiryousefi, Hyvönen & Poczai, 2018a*) from the Solanoideae tribe with a curated plastid genome as an outgroup for rooting our phylogenetic tree. For the species included in our analysis, coding alignments were constructed from the excised plastid genes using MACSE (*Ranwez et al., 2011*), including 75 protein-coding genes (Table S2). For phylogenetic analysis, a 75,449-bp concatenated matrix was used with the best fitting GY+F+I+G4 model determined by ModelFinder (*Kalyaanamoorthy et al., 2017*) according to the Akaike information criterion (AIC), and Bayesian information criterion (BIC). Maximum likelihood (ML) analyses were performed with IQ-TREE (*Nguyen et al., 2015*) using the ultrafast bootstrap approximation (UFBoot; *Hoang et al., 2018*) with 1,000 replicates. The key idea behind UFBoot is to keep trees encountered during the ML-tree search for the original sequence alignment and to use them to evaluate the tree likelihoods for the bootstrap sequence alignment. UFBoot provides relatively unbiased bootstrap estimates under mild model misspecifications and reduces computing time while achieving more unbiased branch supports than standard bootstrap (*Hoang et al., 2018*). TreeDyn was used for further enhancement of the phylogenetic tree (*Dereeper et al., 2008*; *Lemoine et al., 2019*).

Relative divergence times were estimated for species *N. rustica* and putative parental species using BEAST v.1.8.4 (*Drummond et al., 2012*), applying GTR + I + G rate substitution to the protein-coding plastid gene matrix. A Yule speciation tree prior and a uncorrelated relaxed clock model that allows rates to vary independently along branches (*Drummond et al., 2006*) were used, with all other parameters set to default. The median time split between *S. dulcamara* and *N. undulata* (mean = 25 Myr; standard deviation = 0.5) was used as a temporal constraint to calibrate the BEAST analyses derived from the Solanaceae-wide phylogeny of *Särkinen et al. (2013)*. Uncertainty regarding this date was incorporated by assigning normal prior distributions to the calibration point (*Couvreur et al., 2008*; *Evans et al., 2014*). Four independent BEAST runs were conducted with Markov Chain Monte Carlo (MCMC) samples based on 10 million generations, sampling every 10,000 generations. Convergence of all parameters was assessed in Tracer 1.5 (*Rambaut et al., 2014*) and 10% of each chain was removed as burn-in. The Markov chains were combined in LogCombiner 1.7.2. (*Drummond et al., 2012*) to calculate the maximum clade credibility tree.

We defined six biogeographical areas based on the Köppen-Geiger climate classification and further biogeographic evidence and distributions: (A) Colombian/Ecuadorian mountain range mixed equatorial (Af), monsoon (Am), and temperate oceanic climate (Cfb); (B) Northern Peruvian mountain range with tropical savanna climate (Aw); (C) Central Peru with equatorial climate (Af); (D) Coastal Peru with cold semi-arid and desert climate (Bsk, BWk); (E) Peruvian Mountain range with humid subtropical/oceanic highland climate (Cwb); and (F) Bolivian/Chilean alpine/mountain range with mixed semi-arid cold (Bsk, BWk) and humid subtropical climate (Cwa). These areas were used in the Bayesian Binary Method (BBM) model implemented in RASP (*Yu, Blair & He, 2020*) to investigate the biogeographic history of the selected four *Nicotiana* species. BBM infers ancestral area using a full hierarchical Bayesian approach and hypothesizes a special "null distribution", meaning that an ancestral range contains none of the unit areas (*Ronquist, 2004*). The analysis was performed on the BEAST maximum clade credibility tree using default settings, i.e., fixed JC + G (Jukes-Cantor + Gamma) with null root distribution. Ancestral area reconstruction for each node was manually plotted on the BEAST tree using pie charts. Species distributions were determined from data stored in the Solanaceae Source Database (http://solanaceaesource.org/) and Global Biodiversity Information Facility (GBIF) (https://www.gbif.org/).

# RESULTS

## Characteristics of *Nicotiana* plastid genomes

The genome size of the assembled complete plastid genomes ranged between 155,689 bp (*N. paniculata*) and 156,022 bp (*N. obtusifolia*), while reads provided 327 to 1,951x coverage (Table S1). Genomes harbored 133 unique genes, of which 18 genes were duplicated in the IR region (Table S2, Fig. 1A). Out of 133 genes, 85 were protein-coding, 37 were tRNA genes and 8 were rRNA genes. Among 18 duplicated genes in the IR region, 7 were protein-coding, 7 were tRNA genes and 4 were rRNA genes. From the protein-coding genes, 18 contained introns, while *rps* 12 was a trans-spliced gene with its 1st exon found in the LSC and the 2nd and 3rd exons found in the IR region. Structural elements of the IR region also showed the highest GC content (43.2%) compared to the LSC (35.9%) and SSC (32.1%) (Table S1). This finding could be attributed to the presence of tRNA (52.9%) and rRNA (55.4%) genes in inverted repeats.

The nucleotide composition comparison of *Nicotiana* genomes revealed high synteny among all regions, including not only the LSC, SSC, IR and CDSs, but interestingly also in non-coding regions. Detailed comparison of the base composition of each region is shown in Table S3. All amino acid sequences in *Nicotiana* plastid genomes were rich in AT bases and coded a higher percentage of hydrophobic amino acids compared to acidic amino acids (Fig. 1B). Codon usage and frequency of amino acids revealed that leucine is the most abundant and cysteine the least encoded amino acid in these genomes (Fig. 1B). At the 3rd codon position the frequency of A/T codons was higher compared to C/G (Table S4).

The number of RNA editing sites predicted using PREP-cp varied between 34 and 37, distributed among 15 genes (see Table S3). From these genes, the most RNA-edited
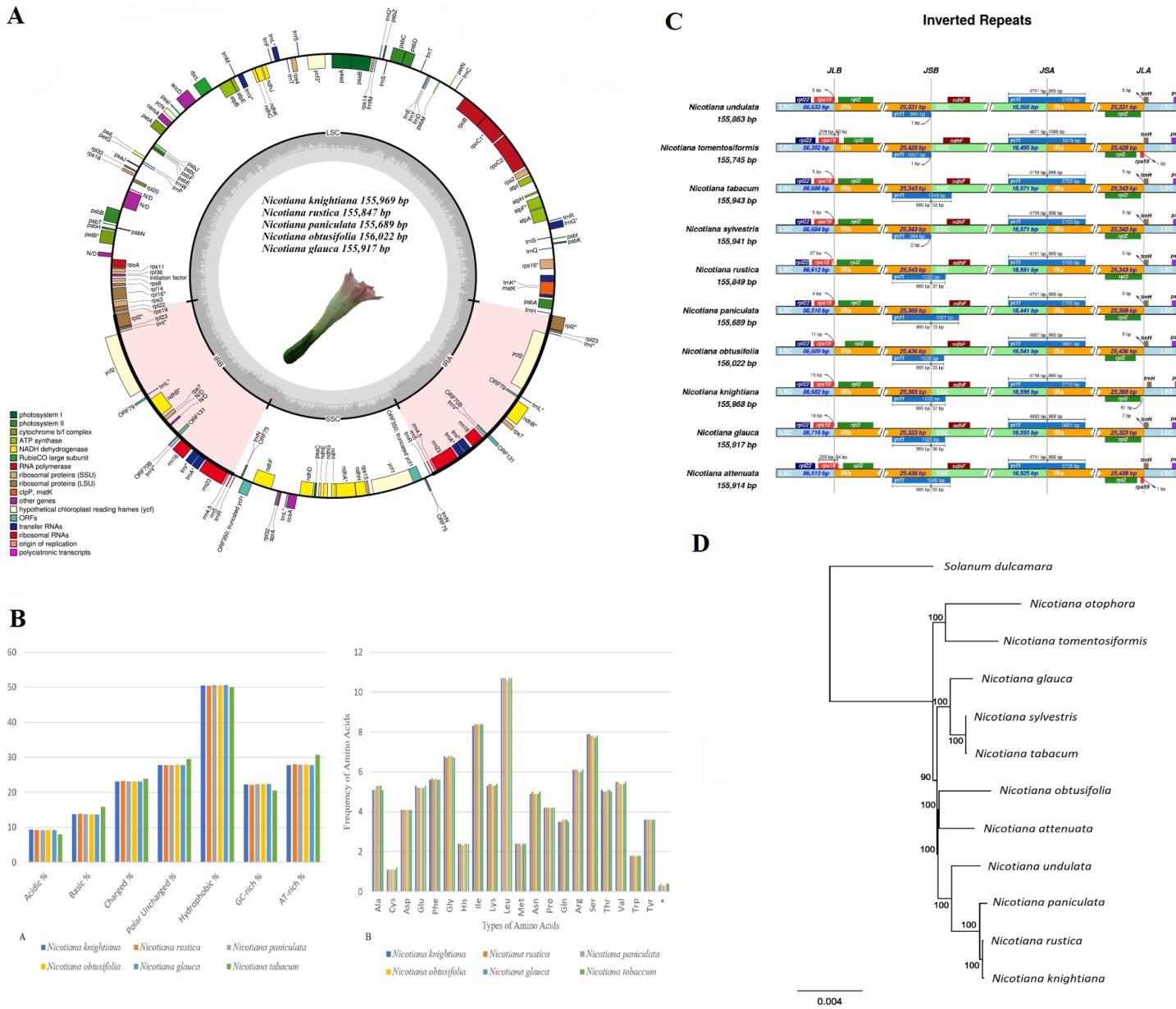

**Figure 1** **Chloroplast genome map of five *Nicotiana* species.** (A) Chloroplast genome map of five *Nicotiana* species. Genes that lie outside the circle are transcribed clockwise while the genes that transcribed counterclockwise are inside the circle. Different colors indicate the genes belonging to various functional groups. GC and AT content of genome are plotted light grey and dark, respectively, in the inner circle. Large single copy (LSC), inverted repeat A (IRa), inverted repeat B (IRb) and small single copy (SSC) are shown in the circular diagram. Inverted repeat regions are highlighted with *cinderella* color. Genes with introns are marked with asterisks. (B) Comparison of amino acid groups and amino acid frequencies in six *Nicotiana* species. (C) Comparison of the border positions of LSC, SSC and IR among the five *Nicotiana* chloroplast genomes. Positive strand transcribed genes are indicated under the line while the genes that are transcribed by negative strands are indicated above the line. Gene names are expressed in boxes, and the lengths of relative regions are showed above the boxes. The number of bp (base pairs) that are written with genes reveal the part of the genes that exists in the region of chloroplast or away from region of chloroplast i.e., bp written with *ycf1* indicate that sequences exist in that region of the plastid genome. (D) Maximum likelihood (ML) tree was reconstructed based on seventy-five protein coding plastid genes of eleven *Nicotiana* species and *Solanum dulcamara* as an outgroup. Bootstrap support values are shown above or below the nodes.

sites were possessed by *ndh*B (9), followed by *ndh*D (6-8) and *rpo* B (4). The *ndh*D gene revealed a fraction of variation among species: *N. knightiana, N. rustica* and *N. paniculata* having six RNA editing sites, whereas seven were observed in *N. obtusifolia* and eight in *N. glauca*. Most of the RNA editing sites were C to U edits on the first and second base of the codons, with the frequency of second base codon edits being much higher. These changes helped in the formation of hydrophobic amino acids, for example valine (V), leucine (L) and phenylalanine (F), with conversions from serine to leucine being the most frequent. (Table S5).

## IR contraction and expansion

The JL (LSC/IR) and JS (IR/SSC) border positions of *Nicotiana* plastid genomes were compared (Fig. 1C) using IRscope (*Amiryousefi, Hyvönen & Poczai, 2018b*). The length of the IR regions was similar, ranging from 25,331 bp to 25,436 bp, with some expansion. The JLA (IRa/LSC) junction point was located between *trnH-GUG* and *rpl2* among *Nicotiana* plastid genomes. In *N. tomentosiformis* and *N. attenuata*, the IR expanded to partially include *rps19*, creating a 60 and 54-bp truncated pseudogenic *rps19* copy at JLA (IRa/LSC). Furthermore, *infA*, *ycf15*, and a copy of *ycf1* located on the JSB (IRb/SSC) were detected as pseudogenes. The position of *ycf* 1 in the IRb/SSC region varied. It left a 33-bp pseudogene in *N. obtusifolia*, a 36-bp pseudogene in *N. knightiana*, *N. rustica* and *N. glauca* and a 72-bp pseudogene in *N. paniculata*.

## Non-synonymous ($K_a$) and synonymous ($K_s$) substitution rate analysis

The synonymous/non-synonymous substitution rate ratio is widely used as an indicator of adaptive evolution or positive selection (*Kimura, 1979*). We have calculated the $K_s$, $K_a$ and $K_a/K_s$ ratio for 77 protein-coding genes for five selected *Nicotiana* species using *N. tabacum* as a reference. Among the analyzed genes, 31 had $K_s = 0$, 19 had $K_a = 0$, and 39 genes had both $K_s$ and $K_a = 0$ values. Of the investigated genes, 13 genes showed $K_a/K_s > 1$ in at least one species (Table S4). We selected these genes for further analysis using FUBAR and MEME. FUBAR estimates the number of nonsynonymous and synonymous substitutions at each codon given a phylogeny, and provides the posterior probability of every codon belonging to a set of classes of $\omega$ (including $\omega = 1$, $\omega < 1$ or $\omega > 1$) (*Murrell et al., 2013*). MEME estimates the probability for a codon to have undergone episodes of positive evolution, allowing the $\omega$ ratio distribution to vary across codons and branches in the phylogeny. This last attribute allows identification of the proportion of codons that may have been evolving neutrally or under purifying selection, while the remaining codons can also evolve under positive selection (*Murrell et al., 2012*). The two models indicated positive selection on the codons only found in *atp*B, *ndh*D, *ndh*F and *rpo*A (Table 1). Thus, the methods described suggested six amino acid replacements altogether as candidates for positive selection, of which three were fixed in all *Nicotiana*, and three were restricted to diverse groups of species (see Table 1).

**Table 1  List of amino acid replacements and results of positive selection tests on codons underlying these replacements.**

| Gene | Position | $\alpha$ | $\beta$ | Amino acid replacements | | | | | | | | | | | FUBAR (PP) | MEME (LRT) | FDR |
|------|----------|----------|---------|-----|------|------|------|------|------|------|------|------|------|------|------------|------------|-----|
| | | | | *Nat* | *Ngla* | *Nkni* | *Nobt* | *Noto* | *Npan* | *Nrus* | *Nsyl* | *Ntab* | *Ntom* | *Nund* | | | |
| *atpB* | 19 | 0.909 | 6.339 | K | N | N | K | K | N | N | N | N | N | N | 0.918 | 4.42 | 0.044 |
| | 21 | 0.697 | 15.511 | P | P | P | P | P | P | P | P | P | P | P | 0.990 | 4.99 | 0.012 |
| *ndhD* | 153 | 1.981 | 12.245 | C | C | C | C | C | C | C | C | C | C | C | 0.927 | 3.16 | 0.021 |
| | 185 | 0.744 | 9.365 | V | L | V | V | V | V | V | L | L | V | V | 0.956 | 3.82 | 0.017 |
| *ndhF* | 460 | 1.385 | 8.772 | V | V | V | V | V | V | V | A | A | V | V | 0.912 | 3.30 | 0.010 |
| *rpoA* | 201 | 1.149 | 8.426 | L | L | L | L | L | L | L | L | L | L | L | 0.940 | 3.84 | 0.023 |

**Notes.**

$\alpha$, the mean synonymous substitution rate at a site; $\beta$, the mean non-synonymous substitution rate at a site;  PP,  posterior probability of positive selection at a site;  LRT,  Likelihood ratio test for episodic (positive) diversification;  FDR,  false discovery rate (FDR = 5%).

Nat, *N. attenuata*; Ngla, *N. glauca*; Nkni, *N. knightiana*; Nobt, *N. obtusifolia*; Noto, *N. otophora*; Npan, *N. paniculata*; Nrus, *N. rustica*; Nsyl, *N. sylvestris*; Ntab, *N. tabacum*; Ntom, *N. tomentosiformis*; Nund, *N. undulata*.

## Repetitive sequences in novel *Nicotiana* plastid genomes

Repeat analysis performed with MISA revealed high similarity in chloroplast microsatellites (cpSSRs) ranging from 368 to 384 among the species. The majority of the SSRs in these plastid genomes were mononucleotide rather than trinucleotide or dinucleotide repeats. The most dominant of the SSRs were mononucleotide A/T motifs, while the second most predominant were dinucleotide AT/TA motifs. Mononucleotide SSRs varied from 7- to 17-unit repeats, dinucleotide SSRs varied from 4- to 5-unit repeats, and other SSRs types present mainly in 3-unit repeats. Most SSRs were located in the LSC and were less frequent in the IR and SSC (Fig. S1 and Table S7). Locating repeats with REPuter revealed 117 oligonucleotide repeats evenly dispersed among the species, ranging from 21 (*N. paniculata*) to 25 (*N. knightiana* and *N. glauca*). Forward (F) and palindromic (P) repeats were abundant in all species, where *N. glauca* had the lowest number of repeats [9 (39%) (F) and 11 (52%) (P)] and *N. obtusifolia* harbored the highest number of repeats [14 (56%) (F) and 11 (44%) (P)]. The size of oligonucleotide repeats varied from 30–65 bp, and many were found in the intergenomic spacer regions (IGS) of the LSC (Fig. S2 and Table S8). The non-coding IGS regions also contained most of the tandem repeats (Fig. S3).

## Single nucleotide polymorphism and insertion/deletion analyses in *Nicotiana*

To discover polymorphic regions (mutational hotspots), the CDS, intron and IGS regions of the whole plastid genome of five *Nicotiana* species were compared. Nucleotide diversity values varied from 0 (*ycf3*) to 0.306 (*rps12* intron region) (Fig. 2). High polymorphism was found in intronic regions (average $\pi = 0.1670$) compared to IGS ($\pi = 0.031$) and CDS regions (average $\pi = 0.002$). We further investigated the number and occurrence of substitution types in the plastid genomes using *N. tabacum* as a reference and encountered 509 (*N. galuca*) to 861 (*N. paniculata*) substitutions along the entire plastid genome. Most of the conversions were A/G and C/T single nucleotide polymorphisms (SNPs) (Table 2). A detailed description of the ratio of transition to transversion substitutions (Ts/Tv) can be found in Table S9. In addition to the distribution of SNPs, we examined insertions and deletions (indels) and located 107 (*N. rustica*) to 143 (*N. obtusifolia*) polymorphisms across the compared genomes (Table 3). Based on this comparison we successfully determined 20 highly polymorphic regions that might be used as potential markers in *Nicotiana* species barcoding (Table 4).

## Phylogenetic analyses

Phylogenetic analysis with *Nicotiana* plastid genomes was performed with the maximum likelihood method based on 75 selected and concatenated protein-coding genes. Our phylogenetic analyses resulted in a highly resolved tree (Fig. 1D). Almost all the recovered clades had maximum branch support values reconstructed based on alignment size of 75,449 bp with best fit model GY+F+I+G4. We further concentrated on the species phylogeny of *N. rustica* and putative parental species, where relative divergence times were estimated using a relaxed uncorrelated clock implemented in BEAST. This analysis found that the divergence of *N. undulata* appeared 5.36 million years ago (Ma) (highest posterior

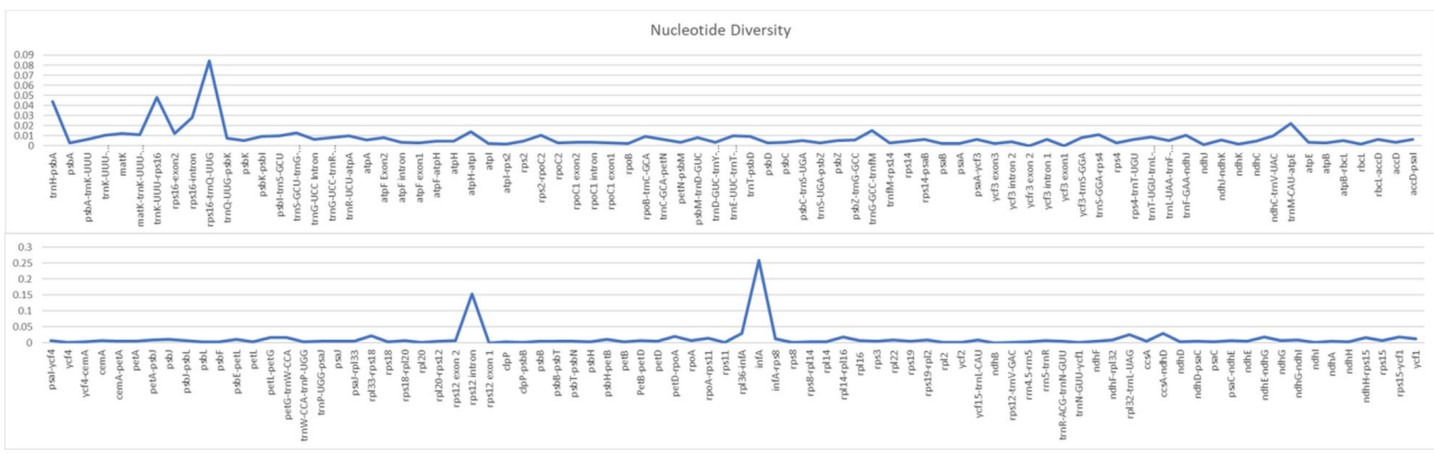

**Figure 2  Nucleotide diversity of various regions of the chloroplast genome among *Nicotiana* species.** The x-axis indicates the chloroplast regions and the y-axis indicates the nucleotide diversity.

**Table 2  Comparison of substitution in *Nicotiana* species.**

| Types | *Nicotiana knightiana* | *Nicotiana rustica* | *Nicotiana paniculata* | *Nicotiana obtusifolia* | *Nicotiana glauca* |
|---|---|---|---|---|---|
| A/G | 222 | 219 | 245 | 244 | 110 |
| C/T | 226 | 223 | 237 | 250 | 128 |
| A/C | 105 | 104 | 117 | 153 | 97 |
| C/G | 40 | 39 | 50 | 52 | 34 |
| G/T | 130 | 129 | 135 | 148 | 87 |
| A/T | 63 | 61 | 77 | 102 | 53 |
| Total | 786 | 775 | 861 | 847 | 509 |
| Location wise distribution | | | | | |
| **LSC** | 560 | 559 | 630 | 671 | 327 |
| **SSC** | 183 | 184 | 198 | 210 | 100 |
| **IR** | 43 | 32 | 33 | 68 | 82 |

**Notes.**
*Nicotiana tabacum* was used as reference for SNPs detection.

density, HPD 6.38–4.43), while *N. paniculata* diverged 1.17 Ma (HPD 2.18–0.63) followed by the most recent split of *N. rustica* and *N. knightiana* 0.56 Ma (HPD 0.65–0.46). This analysis showed that the *Nicotiana* species included in the analysis are not older than the end of the Pliocene and that most subsequent evolution must have occurred in the Pleistocene. The timing of these lineage splits, in addition to the current distributions of four closely related species, were used to infer the progression of migratory steps in RASP (Fig. 3). The most recent common ancestor (MRCA) area illustrated a dispersal event for *N. paniculata* in Northern (B) and Southern Peru (E) and the vicariance of *N. knightiana* in Coastal Peru (D). The overall dispersal pattern of the examined species showed a south-to-north expansion pattern from Central Peru to Colombia and Ecuador (*N. rustica*) to Bolivia (*N. undulata*).

**Table 3   Distribution of InDels in *Nicotiana* chloroplast genome.**

|  | *Nicotiana knightiana* | InDel length (bp) | InDel average length |
|---|---|---|---|
| LSC | 91 | 506 | 5.56 |
| SSC | 11 | 36 | 3.27 |
| IR | 8 | 29 | 3.62 |
|  | *Nicotiana rustica* | | |
| LSC | 89 | 478 | 5.37 |
| SSC | 11 | 36 | 3.27 |
| IR | 7 | 38 | 5.42 |
|  | *Nicotiana paniculata* | | |
| LSC | 92 | 618 | 6.71 |
| SSC | 14 | 156 | 11.14 |
| IR | 10 | 28 | 2.80 |
|  | *Nicotiana obtusifolia* | | |
| LSC | 117 | 677 | 5.78 |
| SSC | 12 | 52 | 4.33 |
| IR | 14 | 167 | 11.92 |
|  | *Nicotiana glauca* | | |
| LSC | 88 | 450 | 5.11 |
| SSC | 11 | 44 | 4 |
| IR | 14 | 82 | 5.85 |

# DISCUSSION

## Molecular evolution of *Nicotiana* plastid genomes

We compared plastid genomes from five *Nicotiana* species, which revealed similar genomic features. These comparative analyses produced an insight into the phylogeny and evolution of *Nicotiana* species. The GC content of the novel *Nicotiana* plastid genomes were similar to those previously reported (*Sugiyama et al., 2005*; *Yukawa, Tsudzuki & Sugiura, 2006*); the GC content was high in the IR, which might be a result of the presence of ribosomal RNA (*Qian et al., 2013*; *Cheng et al., 2017*; *Zhao et al., 2018*). The genome organization, gene order and content also showed high similarity and synteny with sequences previously published for *N. slyvestris* and *N. tabacum* (*Sugiyama et al., 2005*; *Yukawa, Tsudzuki & Sugiura, 2006*). This could be attributed to plastid genomes of land plants having a conserved structure but with diversity prevailing at the border position of LSC/SSC/IR. However, examining the IR junction sites of *Nicotiana* species also showed similarities with some variation prevalent in *N. tomentosiformis*, which has 60 bp in the IRb region, while the *rps19* gene is present entirely in the LSC compared to the *N. tabacum* reference. Such fluctuations at the border positions of various regions of the plastid genome might be helpful in determining evolutionary relationships or could be indicators of environmental adaptation of species (*Menezes et al., 2018*). *Liu et al. (2018)* reported that the similarities in the junction regions may be useful for explaining the relationship between species, and that plants with a high level of relatedness show minimal fluctuations in the junctions of

**Table 4** Mutational hotspots among *Nicotiana* species.

| S. No | Region | Nucleotide Diversity | T. No's of mutation | Region Length |
|---|---|---|---|---|
| 1 | *infA* | 0.2594 | 45 | 249 |
| 2 | *rps*12 intron | 0.1527 | 161 | 527 |
| 3 | *rps*16-*trn*Q-UUG | 0.0845 | 225 | 1,266 |
| 4 | *trn*K-UUU-*rps*16 | 0.0483 | 46 | 703 |
| 5 | *trn*H-*psb*A | 0.0438 | 19 | 433 |
| 6 | *rpl*36-*inf*A | 0.0294 | 3 | 116 |
| 7 | *ccs*A-*ndh*D | 0.0287 | 17 | 237 |
| 8 | *rps*16-intron | 0.0278 | 27 | 862 |
| 9 | *rpl*32-*trn*L-UAG | 0.0261 | 61 | 931 |
| 10 | *trn*M-CAU-*atp*E | 0.0224 | 24 | 218 |
| 11 | *rpl*33-*rps*18 | 0.0222 | 20 | 180 |
| 12 | *pet*D-*rpo*A | 0.0198 | 9 | 182 |
| 13 | *rpl*14-*rpl*16 | 0.0184 | 10 | 119 |
| 14 | *ndh*E-*ndh*G | 0.0173 | 7 | 219 |
| 15 | *rps*15-*ycf*1 | 0.0171 | 17 | 385 |
| 16 | *ndh*H-*rps*15 | 0.0166 | 4 | 108 |
| 17 | *pet*G-*trn*W-CCA | 0.0157 | 4 | 127 |
| 18 | *pet*L-*pet*G | 0.0153 | 6 | 182 |
| 19 | *trn*G-GCC-*trnf*M | 0.0152 | 11 | 228 |
| 20 | *rpo*A-*rps*11 | 0.0151 | 2 | 66 |

the plastid genome. In this respect, the high resemblance of the IR junction sites reveals a close relationship of *Nicotiana* species.

Repeats in the plastid genome are useful in evolutionary studies and play a vital role in genome arrangement (*Zhang et al., 2016*). We detected the presence of large amounts of mononucleotide repeats (A/T), and trinucleotide SSRs (ATT/TAA) in the five analyzed species of *Nicotiana*, which may be a result of the A/T-richness of the plastid genome. A similar result was also reported in *N. otophora L.* (*Asaf et al., 2016*). In all the species of *Nicotiana*, the LSC region contained a greater amount of SSRs in comparison to SSC and IR, which has also been demonstrated in other studies of angiosperm plastid genomes (*Asaf et al., 2016*; *Shahzadi et al., 2019*; *Mehmood et al., 2019*; *Yang et al., 2019*). To understand molecular evolution, it is important to analyze nucleotide substitution rates (*Muse & Gaut, 1994*); in plastid genomes LSC and SSC regions are more prone to substitutions and indels, whereas the IRs are more conserved (*Ahmed et al., 2012*; *Abdullah et al., 2019b*). Our results corroborate this finding, indicating the IR region is more conserved, and most of the substitutions occur in the LSC and SSC regions. Similar results have been shown in the plastid genome of yam (*Dioscorea polystachya* Turcz.) (*Cao et al., 2018*).

Divergence hotspot regions of the plastid genome could be used to develop accurate, robust and cost-effective molecular markers for population genetics, species barcoding, and evolution studies (*Ahmed et al., 2013*; *Ahmed, 2014*; *Nguyen et al., 2017*). Previous studies identified several polymorphic loci based on a comparison of plastid genomes, which have provided suitable information for the development of further molecular markers

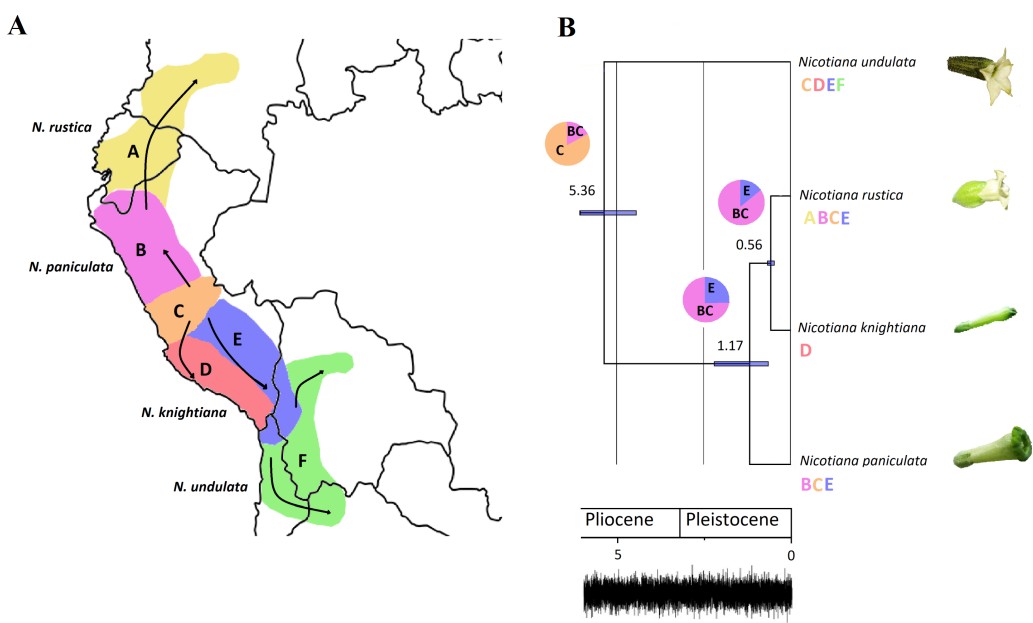

**Figure 3** **Plastome phylogeny and biogeography of the tetraploid *Nicotiana rustica* and related species.** (A) Map showing the six biogeographic areas used to infer the biogeographic history of the *Nicotiana rustica* in South America. Arrows illustrate the dispersal events inferred from the biogeographic analysis. Geographical distribution for each terminal is indicated using the biogeographic regions subdivision. The most probable ancestral area is figured at each node of the phylogenetic tree. Pie-charts represent relative probabilities of ancestral states at each node. (B) Node-calibrated Bayesian maximum clade credibility tree with 95% highest posterior density (HPD) interval for node ages presented as horizontal bars and mean values are displayed above each node. All nodes have PP ≥ 0.97 and BS ≥ 87%. Trace plot of the combined chains showing the sampled joint probability and the convergence of the chains.

(*Choi, Chung & Park, 2016*; *Li et al., 2018*; *Menezes et al., 2018*). We found 20 polymorphism loci in *Nicotiana* that were more polymorphic than the frequently used *rbcL*, and *mat* K markers. For example, *infA*, *rps12* intron and *rps16*-*trnQ*-UUG had nucleotide diversities of 0.2594, 0.1527 and 0.0845, respectively (Table 4).

These regions might show great potential as markers for population genetics and phylogenetic analyses in the genus *Nicotiana*.

## Positive selection on *Nicotiana* plastid genes

Plants have evolved complex physiological and biochemical adaptations to adjust and adapt to different environmental stresses. *Nicotiana,* originating in South America, has spread to many regions of the world and members of the genus have successfully adapted to survive in harsh environmental conditions. This large variation in their distributional range has induced distinctive habits and morphology in inflorescence and flowers, indicative of the physiological specialization to the area where they evolved. Desert ephemeral *Nicotiana* species are short while subtropical perennials have tall and robust habits with variable inflorescences ranging from pleiochasial cymes to solitary flowers and diffuse panculate-cymose mixtures. For example, members of *Nicotiana* section *Suaveolentes*, evolving in

isolation, faced several cycles of harsh climate change. In Australia, the native range of the species, a predominantly warm and wet environment went through intensive aridification (*Poczai, Hyvönen & Symon, 2011*). Throughout this climate change and increasing central aridification, many species either retreated to the wetter coastline or adapted to and still survive in this hostile inland environment (*Bally et al., 2018*). Tobacco plants also developed specialized biosynthetic pathways and metabolites, such as nicotine, which serve complex functions for ecological adaptations to biotic and abiotic stresses, most importantly serving as a defense mechanism against herbivores (*Xu et al., 2017*). *Nicotiana* is thus a rich reservoir of genetic resources for evolutionary biological research since several members of the genus have gone through changing climatic events and adapted to environmental fluctuations.

The patterns of synonymous ($K_s$) and non-synonymous ($K_a$) substitution of nucleotides are essential markers in evolutionary genetics defining slow and fast evolving genes (*Kimura, 2006*). $K_a/K_s$ values $>1$, $=1$, and $<1$ indicate positive selection, neutral evolution and purifying selection, respectively (*Lawrie et al., 2013*). Many proteins and RNA molecules encoded by plastid genomes have undergone purifying selection since they are involved in important functions of plant metabolism, self-replication, and photosynthesis and therefore play a pivotal role in plant survival (*Piot et al., 2018*). Departure from the main purifying selection in the case of plastid genes might happen in response to certain environmental changes when advantageous genetic mutations can contribute to better survival and adaptation. The $K_a/K_s$ ratios in our analysis indicate changes in selective pressures. The genes *atp*B, *ndh*D, *ndh*F and *rpo*A had greater $K_a/K_s$ values ($>1$), possibly due to positive selective pressure as a result of specific environmental conditions. This was conclusively supported by an integrative analysis using Fast Unconstrained Bayesian AppRoximation (FUBAR) and Mixed Effects Model of Evolution (MEME) methods, which identified a set of positively selected codons in these genes.

These genes are involved in different plastid functions, such as DNA replication (*rpo*A) and photosynthesis (*atp*B, *ndh*D and *ndh*F). The *rpo*A gene encodes the alpha subunit of PEP, which is believed to predominantly transcribe photosynthesis genes (*Hajdukiewicz, Allison & Maliga, 1997*). The transcripts of plastid genes encoding the PEP core subunits are transiently accumulated during leaf development (*Kusumi et al., 2011*), thus the entire *rpo*A polycistron is essential for chloroplast gene expression and plant development (*Zhang et al., 2018*). The housekeeping gene *atp*B encodes the $\beta$-subunit of the ATP synthase complex, which has a highly conserved structure that couples proton translocation across membranes with the synthesis of ATP (*Gatenby, Rothstein & Nomura, 1989*), which is the main source of energy for the functioning of plant cells. In chloroplasts, linear electron transport mediated by PSII and PSI produces both ATP and NADPH, whereas PSI cyclic electron transport preferentially contributes to ATP synthesis without the accumulation of NADPH (*Peng & Shikanai, 2011*). Chloroplast NDH monomers are sensitive to high light stress, suggesting that the *ndh* genes encoding NAD(P)H dehydrogenase (NDH) may also be involved in stress acclimation through the optimization of photosynthesis (*Casano, Martín & Sabater, 2001*; *Martin et al., 2002*; *Rumeau, Peltier & Cournac, 2007*). During acclimation to different light environments, many plants change biochemical

composition and morphology (*Terashima et al., 2005*). The highly responsive regulatory system controlled by cyclic electron transport around PSI could optimize photosynthesis and plant growth under naturally fluctuating light (*Yamori, 2016*). When demand for ATP is higher than for NADPH (e.g., during photosynthetic induction, at high or low temperatures, at low $CO_2$ concentration, or under drought), cyclic electron transport around PSI is likely to be activated (*Yamori, 2016*; *Yamori & Shikanai, 2016*). Thus, positive selection acting on ATP synthase and NAD(P)H dehydrogenase encoding genes is probably evidence for adaptation to novel ecological conditions in *Nicotiana*.

These findings are further supported by our observation that RNA editing sites occur frequently in *Nicotiana ndh* genes (Table S3). It has been shown that *ndh*B mutants under lower air humidity conditions or following exposure to ABA present reduced levels of photosynthesis, likely mediated through stomatal closure triggered under these conditions (*Horvath et al., 2000*). Therefore, a protein structure modification resulting from a loss or decrease in RNA editing events could affect adaptations to stress conditions or cause other unknown changes (*Rodrigues et al., 2017*). Previous studies have demonstrated that abiotic stress influences the editing process and consequently plastid physiology (*Nakajima & Mulligan, 2001*). Alterations in editing site patterns resulting from abiotic stress could be associated with susceptibility to photo-oxidative damage (*Rodrigues et al., 2017*) and indicate that *Nicotiana* species experienced abiotic stresses during their evolution, which resulted in positive selection of some plastid genes. Up to this point, positive selection has rarely been detected in plastid genes except for *clp*P (*Erixon & Oxelman, 2008*), *ndh*F (*Peng, Yamamoto & Shikanai, 2011*), *mat*K (*Hao, Chen & Xiao, 2010*) and *rbc*L (*Kapralov et al., 2011*). However, *Piot et al. (2018)* showed that one-third of the plastid genes in 113 species of grasses (Poaceae) evolved under positive selection. This indicates that positive selection is overlooked among diverse groups of plant taxa.

## Phylogenetic relationships and the origin of tetraploid *Nicotiana rustica*

Our comparative plastid genome analysis revealed that the maternal parent of the tetraploid *N. rustica* was the common ancestor of *N. paniculata* and *N. knightiana*, with the latter species being more closely related to *N. rustica*. The relaxed molecular clock analyses estimated that the speciation event between *N. rustica* and *N. knightiana* appeared ~0.56 Ma (HPD 0.65–0.46), in line with previous findings (*Sierro et al., 2018*). Comparative analysis of the genomes of four related *Nicotiana* species revealed that *N. rustica* inherited about 41% of its nuclear genome from its paternal progenitor, *N. undulata*, and the rest from its maternal progenitor, the common ancestor of *N. paniculata* and *N. knightiana* (*Sierro et al., 2018*), which has also been confirmed by our study. We also revealed that *N. knightiana* is more closely related to *N. rustica* than *N. paniculata*, which can be further corroborated by the distribution of indels highlighted in the present study. The paternal inheritance of plastid genomes was observed in *Nicotiana* under certain stressed conditions (*Medgyesy, Fejes & Maliga, 1985*; *Medgyesy, Páy & Márton, 1986*; *Thang & Medgyesy, 1989*; *Avni & Edelman, 1991*; *Ruf, Karcher & Bock, 2007*; *Thyssen, Svab & Maliga, 2012*;) Such low-frequency paternal leakage of plastids via pollen was suggested to be universal in plants

with strict maternal plastid inheritance (Azagiri & Maliga, 2007). Thus, we expect that the plastids in the putative parents of *N. rustica* are maternally inherited. Medgyesy, Páy & Márton (1986) observed the paternal transmission of plastids in *N. plumbaginifolia* Viv., but they concluded that plants carried maternal mitochondria. Further studies investigating the parental origins of *Nicotiana* species should also focus on mitochondrial genomes excluding possible low-frequency paternal plastid inheritance.

The biogeographical analysis suggests that *N. undulata* and *N. paniculata* evolved in North/Central Peru, while *N. rustica* developed in Southern Peru and separated from *N. knightiana,* which adapted to the Southern coastal climatic regimes. Positively selected plastid genes with functions such as DNA replication (*rpo*A) and photosynthesis (*atp*B, *ndh*D and *ndh*F) might have been associated with successful adaptation to, for example, a coastal environment. However, our results are tentative, as our study lacks data for several broad ecological variables, including variation in salinity, island versus mainland, and East versus West of the Andes. We aim to highlight that many potential environmental variables might be highly correlated with speciation processes in *Nicotiana*, as has been demonstrated in the same region for another Solanaceae group in the tomato clade (*Solanum* sect. *Lycopersicon*), where amino acid differences in genes associated with seasonal climate variation and intensity of photosynthetically active radiation have been correlated with speciation processes (*Pease et al., 2016*). Another example of rapid adaptive radiation from the family is the genus *Nolana* L.f., where several clades gained competitive advantages in water-dependent environments by succeeding and diverging in Peru and Northern Chile (*Dillon et al., 2009*). In the case of *N. rustica* and related species, we assume that diversification was driven by the ecologically variable environments of the Andes. Our molecular clock analysis provides evidence for recent species diversification in the Pleistocene and Pliocene while substantial climatic transitions in Peru predate these events. For example, the uplift of the central region of the Andes and the formation of the Peruvian coastal desert ended (*Hoorn et al., 2010*; *Gerreaud, Molina & Farias, 2010*), before the geographical and ecological expansion of *N. rustica* and related parental species.

The dispersal of *N. rustica* and related species shows a south-to-north range expansion and diversification which has been suggested by phylogenies of other plant and animal groups in the Central Andes (*Picard, Sempere & Plantard, 2008*; *Luebert & Weigend, 2014*). Based on the south-to-north progression scenario, habitats located at high altitudes were first available for colonization in the south, recently continuing to northward. Erosion and orogenic progression caused dispersal barriers of species colonizing these high habitats to diversify in a south-to-north pattern, frequently following allopatric speciation. Thus, for taxonomic groups currently residing throughout a large portion of the high Andes, a south-to-north speciation pattern is expected (*Doan, 2003*). In this case, the most basal species (*N. undulata*) has more southern geographic ranges, and the most derived species (*N. rustica*) has more northern geographic ranges, except for *N. knightiana*, which presumably colonized the coastal range of Peru. Although the four *Nicotiana* species examined show overlaps in their distribution, it is probable that speciation was caused by fragmentation of populations during the glacial period (see *Simpson, 1975*). Utilizing fewer chloroplast loci for phylogenetic analyses of plant species may limit the solution of phylogenetic

relationships, specifically at low taxonomic levels (*Hilu & Alice, 2001*; *Majure et al., 2012*). Previously, *Nicotiana* was subdivided into 13 sections using multiple chloroplast markers, i.e., *trn*L intron and *trn*L-F spacer, *trn*S-G spacer and two genes, *ndh*F and *mat*K (*Clarkson et al., 2004*). Recently, inference of phylogeny based on complete plastid genomes has provided deep insight into the phylogeny of certain families and genera (*Henriquez et al., 2014*; *Amiryousefi, Hyvönen & Poczai, 2018a*; *Abdullah et al., 2020*). Here, we reconstructed a phylogenetic tree for eleven species of *Nicotiana* that belong to nine sections (*Clarkson et al., 2004*) based on 75 protein-coding genes by using *S. dulcamara* as an outgroup, which attests the previous classification of genus *Nicotiana* with high bootstrap values. Species of each section are well resolved whereas *N. tabacum* of section *Nicotiana*, and *N. sylvestris* of section *Sylvestres*, show close resemblance. *N. paniculata* and *N. knightiana* belong to section Paniculatae and appeared to reflect the maternal ancestry of these species relative to *N. rustica* of section *Rusticae*. Overall, our phylogenetic analyses support the previous classification of genus *Nicotiana,* and corroborates that plastid genomic resources can provide further support for highly resolved phylogenies.

## CONCLUSION

In the present study, we assembled, annotated and analyzed the whole plastid genome sequence of five *Nicotiana* species. The structure and organization of their plastid genome was similar to those of previously reported Solanaceae plastid genomes. Divergences of LSC, SSC and IR region sequences were identified, as well as the distribution and location of repeat sequences. The identified mutational hotspots could be utilized as potential molecular markers to investigate phylogenetic relationships in the genus, as we demonstrated in our study to elucidate the maternal genome origins of *N. rustica*. Our results could provide further help in understanding the evolutionary history of tobaccos.

## ACKNOWLEDGEMENTS

We thank Kenneth Quek and Kathryn Rannikko for editing the manuscript.

### Funding

The Higher Education Commission (HEC), Pakistan, granted Furrkh Mehmood a scholarship under the International Research Support Initiative Program (IRSIP) to conduct research work at the Botany Unit, Finnish Museum of Natural History, University of Helsinki, Finland. The funder had no role in study design, data collection and analysis, decision to publish, or preparation of the manuscript.

### Competing Interests

Ibrar Ahmed is employed by Alpha Genomics Private Limited. The authors declare there are no competing interests.

## Author Contributions

- Furrukh Mehmood conceived and designed the experiments, performed the experiments, analyzed the data, prepared figures and/or tables, authored or reviewed drafts of the paper, and approved the final draft.
- Abdullah analyzed the data, authored or reviewed drafts of the paper, and approved the final draft.
- Zartasha Ubaid and Iram Shahzadi analyzed the data, authored or reviewed drafts of the paper, and approved the final draft.
- Ibrar Ahmed, Mohammad Tahir Waheed and Bushra Mirza conceived and designed the experiments, authored or reviewed drafts of the paper, and approved the final draft.
- Peter Poczai performed the experiments, analyzed the data, prepared figures and/or tables, authored or reviewed drafts of the paper, and approved the final draft.

## DNA Deposition

The following information was supplied regarding the deposition of DNA sequences:

Plastid genome sequences are available in the Supplemental Files and NCBI: BK010737 (*Nicotiana knightiana*), BK010738 (*Nicotiana rustica*), BK010741 (*Nicotiana paniculata*), BK010739 (*Nicotiana obtusifolia*), and BK010740 (*Nicotiana glauca*).

## Data Availability

Illumina sequence data of *Nicotiana knightiana* L. (SRR8169719), *N. rustica* (SRR8173839), *N. paniculata* (SRR8173256), *N. obtusifolia* (SRR3592445) and *N. glauca* (SRR6320052) available at the Sequence Read Archive (SRA).

## Supplemental Information

Supplemental information for this article can be found online at http://dx.doi.org/10.7717/peerj.9552#supplemental-information.

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
