# Peer review of "Plastid genomics of Nicotiana (Solanaceae): insights into molecular evolution, positive selection and the origin of the maternal genome of Aztec tobacco (Nicotiana rustica)"

_PeerJ, doi:10.7717/peerj.9552_

## Round 0.1 · original submission · Major Revisions

Please go over the comments of the two reviewers, which made excellent reviews. In general readability should be improved, several sentences should be re-phrased. With regard to methods they should be better explained, providing details for example on the program parameters used to carry out analyses. In relation to references, some citations are missing citations and moreover citations are needed to include important Nicotiana background that were not included in Introduction.

Reviewer 1 ·

Basic reporting

The authors perform a study of the plastidial genome of different Nicotiana species, centered on Nicotiana rustica (aztec tobacco). Although the Discussion section is well written, the Introduction and the Results could benefit from a more extended language revision. In more than one case, a sentence was hard to understand or contained errors.
In the Introduction some of the needed citations are either outdated or missing, which results in incorrect dates reported for certain species. Most importantly, there is a missing point that should be addressed: Paternal chloroplast transmission has been observed in Nicotiana species before, under certain stress conditions or reproduction systems. Since Nicotiana is one of few species where this is known it should be mentioned in the text.

Specific comments

41-43: It should be clear for any reader that, besides the maternal parent who is hereby identified, there is a candidate paternal parent (N. undulata) (Sierro et al, 2018 https://doi.org/10.1186/s12864-018-5241-5 ). This point is reported in the Discussion (4.3) but not in the introduction or abstract.

69-70: The smoking tobacco hybridization date and the Suaveolentes hybridization date are currently placed at 0.4 and 5 million years ago, respectively (Clarkson et al., 2017 https://doi.org/10.1007/s00606-017-1416-9 , Schiavinato et al., 2020 https://doi.org/10.1111/tpj.14648 ).

72: “amphidiploidy” (correct: amphidiploid) is used wrongly. N. tabacum cannot be considered an amphi-diploid because it does not contain two entirely conserved diploid genomes within its nucleus, as it would be in the moment of hybridization (Lim et al., 2007 https://doi.org/10.1111/j.1469-8137.2007.02121.x ). Hence, it should be referred to as a “hybrid” or as an “allopolyploid”.

73-74: Not only N. tabacum, but also N. benthamiana is used as model organism, perhaps even more, and comprehensive resources exist for N. benthamiana as well (Goodin et al, 2008 https://doi.org/10.1094/MPMI-21-8-1015 , Bally et al 2018 https://doi.org/10.1146/annurev-phyto-080417-050141 , Schiavinato et al, 2019, https://doi.org/10.1186/s12864-019-5960-2).

92: Paternal inheritance of chloroplast DNA was observed also in Nicotiana under certain conditions of stress or reproduction ( Medgyesy et al, 1986 https://doi.org/10.1007/BF00425497 , Avni et al, 1991 https://doi.org/10.1007/BF00269859 ). Although this doesn't affect the claim that maternal inheritance is to be expected, it should be addressed in the paper.

154: The species name S. dulcamara comes out here for the first time, but it is not clear why it was used. Although it might be clear for the authors, it has to be at least referred to in the methods as well (e.g. “a species from the nightshades used as control”) and a reference should be added where the data were obtained from. I believe that they used it because some of the authors already published the chloroplast sequence of this species (Amiryousefi et al., 2018 https://doi.org/10.1371/journal.pone.0196069 ). Also, the name should be written entirely since it's the first appearance (Solanum dulcamara). This issue is reiterated at line 268, where S. dulcamara is intrudoced as a reference species for the Ka/Ks ratio calculation.

255-256: “The endpoint of the Solanaceae JLA is characteristically located upstream of the rps19 and downstream of the trnH-GUG.” The sentence is copy-pasted from a previous publication of some of the authors (Amiryousefi et al., 2018 https://doi.org/10.1371/journal.pone.0196069 ). Despite that, the JLA term is not defined or referenced, so the reader must rely on personal knowledge to understand it. According to Meng et al (2019) (https://doi.org/10.1371/journal.pone.0211340 ) it is a known chloroplast marker. Although this is common knowledge in the field of plastid genomics, it should be clarified in the text (the previous publication does not report details on it either).

372-375: “The size range of LSC, SSC and IR varies between the plastid genomes of species that advances to alterations...” -- the point expressed by the authors is unclear, please rephrase.

499-500: “N. rustica and in fact all Nicotiana tetraploids, except species included in section Suaveolentes, originated from a doubling of the diploid chromosome for the genus”. This sentence is unclear. The authors do not specify to which chromosome they are referring. In each Nicotiana hybrid, the doubling of the genome (not just one chromosome) happened through hybridization between different Nicotiana species. This is true for section Suaveolentes as well (Kelly et al, 2013 https://doi.org/10.1111/j.1558-5646.2012.01748.x ; Clarkson et al, 2017 https://doi.org/10.1007/s00606-017-1416-9 ; Bally et al 2018 https://doi.org/10.1146/annurev-phyto-080417-050141 ). Another work (Schiavinato et al., 2020 https://doi.org/10.1111/tpj.14648 ) even shows that the lineage of Nicotiana section Suaveolentes is now fully resolved. I believe that it was not available at the time this paper was written; however, the other ones were. Overall, this sentence should be rewritten adjusting the content and including the provided citations.

549-550: “further data is required to elucidate the phylogenetic relationship among these two species”. This sentence can be expanded. The work of Schiavinato et al., 2020 https://doi.org/10.1111/tpj.14648 showed how very large collections of phylogenetic trees can overpower the limitations of using a small set of genes. In fact, previous studies on Nicotiana tetraploid species ( Clarkson et al, 2017 https://doi.org/10.1007/s00606-017-1416-9 ) could not perfectly resolve the lineage of section Suaveolentes (i.e. the oldest Nicotiana hybrids). This was likely due to insufficient signal due to excessive sequence divergence.

Readability and Grammar

28-29: sentence is unclear. From a grammar standpoint, “tobacco plants” refers to Solanaceae, while the authors probably wanted to refer to the Nicotiana genus.

28: “are a group” refers to the Nicotiana genus, hence “is a group”. The authors probably referred to “tobacco plants”. The sentence should be reorganized in terms of grammar.

33: showed “the” typical.

80: specify “Proteins” e.g. “Proteins contained therein”.

102: “transplastomic” should be plural (transplastomics).

151: “substitution” should be plural (substitutions).

168+337: The headings in Methods and Results include the word “phylogenomic” but throughout the paragraph it is used as “phylogenetic”. The latter is probably more accurate, since only the chloroplast genome is analysed.

174-177: To increase readability, a supplementary table would be recommended instead of a list of genes.

177: “a matrix was used” instead of “we used a matrix” (according to the recommended grammar for methods).

200: “Six biogeographical areas were defined” instead of “we defined six biogeographical areas” (according to the recommended grammar for methods).

222-223: The “x” that follows each coverage value means “cross product” (or vectorial product). It is commonly used when two parents of a hybrid are described together, to signify that the hybrid is the product of those. In the context of coverage, an “x” (the letter) should be used.

224: “was highest” instead of “were highest”.

237: “was encoded” instead of “were encoded”, as it refers to percentage (singular).

377: “which has” instead of “which have”.

378: “the rps19 gene” (missing “the”).

569: Author name “Abdullah” without firstname/surname correct?

Experimental design

The Materials and Methods did not include crucial information such as program parameters that would enable other scientists to reproduce the analyses. Please check all programs mentioned in the Methods section if the information given in the text is sufficient to reproduce the results.

Specific comments

114-115: The authors should specify which bwa subprogram was used (i.e. “mem”, “aln” , “fastmap” or “bwasw”, as the result can vary greatly. In fact, given that long queries (contigs) were mapped the alignment strategy should take into account the expected mismatches between contigs and the reference used. This is true especially because the reference chloroplast genome is not from the same species, hence some divergence is expected.

116: missing Geneious parameters

118: missing GeSeq and CPGAVAS2 parameters

119: “manually inspected and curated” please expand with explanation of the strategy used for curation.

121: missing Aragorn parameters

122: missing OGDRAW parameters

124: as for lines 114-115, in this case short reads are used and therefore a different mapping strategy was likely used but is not described

136: missing IRScope parameters

239: “Relative synonyms codon usage (RSCU)” is introduced in the text but not explained. The reader can only rely on personal knowledge to understand this term, since no reference is provided. One reference could be Sharp and Li (1987) (https://doi.org/10.1093/nar/15.3.1281) but the authors should clarify which codon usage bias model they used.

270-274: The reference values indicated for Ka/Ks ratios are somewhat arbitrary (0.5 and 4). As reported in the Discussion, Ka/Ks < 1 means purifying selection while Ka/Ks > 1 means positive selection (Ka/Ks = 1 being the neutral case). When reporting the results, it would be more informative to report the number of genes whose ratio is above, below or equal to 1. In addition to that, the whole sentence could benefit from rearrangement to make it clearer, for example: line 273-274 say “whereas genes rpoC1, atpB, rpoA, ndhD had Ka/Ks ratio more than 4 species”. It is not clear from the sentence whether 4 is the number of species or the Ka/Ks value (until the Supplementary Table S6 is consulted). Later in the paragraph (line 278) the correct ω reference values are reported. More in general, if a rationale is used to choose 0.5 and 4 as reference values, it should be explained in the text.

Validity of the findings

Newly generated/assembled sequence data should be made available in a public database (e.g. NCBI GenBank).

Additional comments

In the Discussion, the authors first address the different structures of plastidial genomes found between the analysed Nicotiana species. Specifically, they show that a series of loci are more variable than others that are commonly used as markers. Hence, the more variable loci could be used as well. This conclusion is interesting and based on convincing information.
They later show results on molecular evolution of plastidial genes, highlighting positive selection in multiple ones. They conduct the study with the right tools and come to interesting conclusions. Specifically, they show how positive selection of plastidial genes is often overlooked, and parallel their results to the ones obtained for more than a hundred grass species. The Discussion of these results is particularly well-written.
Finally, they analyse the phylogenetic relationships that defined the tetraploid hybrid Nicotiana rustica. The discussion of these results requires some rearrangements. As pointed out in the comments above, it is unclear what the authors mean when referring to section Suaveolentes and it could potentially be a wrong claim.

Reviewer 2 ·

Basic reporting

Piont 1. English must be thoroughly revised.

Point 2. Overall manuscript is used word plastome, chloroplast genome, and plastid genome in a mixtures, so you should be consistent. Likewise, unify the words "ψ" and "pseudogene".

Point 3. It would be better to provide candidate primer sequences or multiple alignment sequences for hotspot regions.

Point 4. Manuscript is exist many irrelevant references and duplicates sentences.

Experimental design

Point 1. Briefly explain why you mainly targeted 5 species.

Validity of the findings

Point 1. Ka, Ks, and Ka/Ks values may be more suitable to average value after cross-correlate comparison, rather than comparison with a reference.

Point 2. Figure 4 to 6 seem to like to show as a single figure, or trun it over to supplementary figure.

Point 3.In abstract, N. rustica was the common ancestor of N. paniculata and N. knightiana. However, both N. rustica and N. knightiana was constructed a sister group in figure 8 and 9. Please explain it more clearly.

Additional comments

Line 78. It would better "furthermore" than "while"

Lines 80-81. It would better "Most proteins are functioned for photosynthesis along with" than "Proteins are used not only for photosynthesis but also for".

Line 85. It would better "typically" than "typical".

Line 86. It would better "which are consist of" than "with".

Lines 88-90. Does it mean that structural modifications oucur frequently?

Line 91. It would better "have been known a maternal inheritance" than "have a uniparental maternal inheritance".

Line 92. It would better "reported" than "recorded".

Line 95. It would better "took advantage of" than "exploited".

Lines 100-103. There are duplicated above, change the location of the paragraph and delete the duplication.

Line 111. Change from "Nicotiana knightiana L." to "N. knightiana".

Lines 114-116. BWA is mapping program for reference sequence, not a program for conig selection. Did you used for a reference mappping? Is this right, contents should be modified accordingly.

Lines 116-117. You should provided the parameters used for de novo assembly in more detail.

Line 117. It would better "Gene annotation was conducted" than "The genome sequence was annotated".

Line 118. It would better "After draft (or automatical) annotation ?" than "Following de novo annotation"

Line 119. Explain in more detail about manually inspected and curated.

Line 121. You should provided the parameters of Aragorn.

Line 121. drawn with -> drawn by

Line 123. determined -> calculated

Line 123. Explain the "mapping all reads" correctly. Is it raw reads or trimmed reads?

Lines 135-136. The expansion and contraction of inverted repeats and their border positions -> Structural border of chloroplast genome

Line 136. ten selected Nicotiana species -> 10 Nicotiana species

Line 142. Microsatellites repeats -> Microsatellites

Line 142. detected -> identified

Lines 151-152. It looks like an unnecessary sentence.

Line 177. For phylogenetic analysis -> For phylogenetic analysis, (added comma)

Line 195. As you mentioned 10 million generations, is this meant MCMC chain-length? If correct, it should be modified.

Line 218. I thought "species" would be better to delete.

Line 218. assembled and the lengths of these plastid genomes were: -> completely assembled those genome size as follows:

Lines 222-226. It seems better to write the size and GC contents of each structure rather than average coverage.

Line 227. De novo assembled Nicotiana plastid genomes had 133 unique genes, whereas -> Nicotiana plastid genomes harbored 133 unique genes, of those

Line 254. Add to reference paper of IRscope

Lines 294-296. Is this sentence meant to investigate the dispersed repeat? If this is right, write to purpose of analysis in more datail.

Line 338. What is mean of "reconstructed"?

Line 339. Add to number of concatenated protein-coding genes.

Lines 376-377. Nicotiana tabacum -> N. tabacum
Nicotiana tomentosiformis -> N. tomentosiformis

Line 387. Specify to the exact number of species, not "all the species".

Line 393. Using without abbreviation of Ts/Tv, or paraphrasing to Ka/Ks.

Line 411. polymorphic regions -> polymorphism

Lines 418-434. Those paragraphs are look like the introduction.

Line 437. It would better "neutral selection" than "natural evolution".
You are previously wrote a positive selection, so you should be used "negative selection" than "purifying selection".

Line 485. Is this a typo? clpP1 -> clpP

Line 487. I thought "a recent study by " would be better to delete.

Lines 488-489. Following by "This might ~", what did you think so?

Lines 508-510. It is not good "However, our results should be regarded as tentative ~ as below" for discussion. Sentence should be replaced.

---

## Round 0.2 · Minor Revisions

Please attend this request by one of the reviewers: copy the new or modified sentence(s) along with their answers below each comment to facilitate a second review of the manuscript. Specify line numbers. This will help reviewers to see whether you considered suggestions.

Reviewer 1 ·

Basic reporting

see "general comments"

Experimental design

see "general comments"

Validity of the findings

see "general comments"

Additional comments

I have difficulties to find the modifications since the text has heavily changed overall, and only a small fraction of changes is highlighted in the “grid” version. Can the authors please copy the new or modified sentence(s) along with their answers below each comment to facilitate a second review of the manuscript. Specifying line numbers would be helpful, too. Thanks.

---

## Round 0.3 · Minor Revisions

i appreciate your consideration of every issue raised by the two reviewers, it improved significantly the manuscript. They were many points and reading your Rebuttal letter you took every one seriously. I just have these particular issues: for the positive selection results (table 1) the false discovery rate (FDR) values from FUBAR and MEME need to be given. FDR should be used for significance threshold, not the PP values. Also, line 198 "sites with PP < 0.9". Shouldn't it be "PP > 0.9"? Could you please consider them?

---

## Round 0.4 · accepted · Accept

Thank you for adding FDR values in Table 1 and the reference. I found a typo on the type of inflorescence that you can correct in proofs. Perhaps, when you review the proofs you can add a map of South America, very small, to better understand the marked Pacific area where these Nicotiana studied species are distributed.